# Esketamine Inhibits Cocaine-Seeking Behaviour Subsequent to Various Abstinence Conditions in Rats

**DOI:** 10.3390/biom13091411

**Published:** 2023-09-19

**Authors:** Karolina Wydra, Kacper Witek, Agata Suder, Małgorzata Filip

**Affiliations:** Department of Drug Addiction Pharmacology, Maj Institute of Pharmacology, Polish Academy of Sciences, 12 Smętna Street, PL-31-343 Kraków, Poland; witek@if-pan.krakow.pl (K.W.); suder@if-pan.krakow.pl (A.S.)

**Keywords:** cocaine abstinence, cocaine-seeking behaviour, esketamine, rats

## Abstract

Background: Cocaine use disorder (CUD) is a relapsing brain disease caused by a chronic drug intake that involves neural mechanisms and psychological processes, including depression. Preclinical and clinical studies have demonstrated the promise of pharmacological drugs in controlling the reinstatement of cocaine by targeting the N-methyl-D-aspartate (NMDA) receptor. Recent evidence has revealed that esketamine, a (S) enantiomer of ketamine, shows a high affinity to NMDA receptors and has been used in clinical trials to treat moderate-to-severe depression. Methods: In the present paper, we investigated the effects of esketamine in regulating cocaine-seeking behaviour induced through the use of cocaine (10 mg/kg) or the cocaine-associated conditioned cue after a short (10 days)-lasting period of drug abstinence with extinction training, home cage or enrichment environment conditions in male rats. Furthermore, we investigated the acute effects of esketamine on locomotor activity in drug-naïve animals. Results: Esketamine (2.5–10 mg/kg) administered peripherally attenuated the reinstatement induced with cocaine priming or the drug-associated conditioned cue after different conditions of abstinence. Conclusions: These results seem to support esketamine as a candidate for the pharmacological management of cocaine-seeking and relapse prevention; however, further preclinical and clinical research is needed to better clarify esketamine’s actions in CUD.

## 1. Introduction

Cocaine use disorder (CUD) is a brain disorder with a multifactorial pathology, having somatic, psychiatric, cognitive and social consequences [1]. The disorder is associated with the dysregulation of multiple neuronal systems, such as dopaminergic (DA) transmission, within the brain’s reward system [2,3]. The findings also support the strong impact of glutamatergic (Glu) signalling over cocaine seeking and relapse [3].

One promising Glu drug in animal and human studies is ketamine, a racemic mixture containing (R) and (S) enantiomers in equal parts [4,5]. In human studies, the (S) enantiomer called esketamine is a more potent anaesthetic and analgesic than (R)-ketamine [6], while, in animal research, it also has been shown to have a 3–4 times higher affinity for NMDA receptors as compared to (R)-ketamine (Ki = 0.30 μM and Ki = 1.4 μM, respectively) [7]. Esketamine has shown antidepressant effects in preclinical research [8,9] and clinical trials [10,11]. Based on these observations, an intranasal delivery of esketamine (branded as Spravato^®^) was recently approved in the United States and Europe to treat treatment-resistant depression. Other findings indicate anxiolytic-like effects of esketamine [12].

Similar to ketamine, esketamine increases the prefrontal DA and serotonin release [7] and reduced striatal DA D2/3 receptor binding in rodents [13]. Such a neurochemical profile may indicate esketamine’s psychotomimetic effects; however, a single clinical report shows a lack of psychotic symptoms [14,15].

In the present behavioural and pharmacological analyses, we examined the effects of esketamine on cocaine-seeking behaviour following several abstinence conditions in male rats. Esketamine was chosen since it possesses mood-enhancing properties that partially tie in with the reduction in stress conditions during drug abstinence that have significant implications as a trigger factor capable of restoring cocaine seeking. In fact, the greatest difficulty in treating CUD is the enormous rates of relapse, which depend also on the abstinence conditions that may affect the intensification of relapse. Thus, social isolation as a negative environmental issue rises [16,17,18], while an enriched environment or behavioural cue extinction therapy serving as positive environmental issues reduce the risks of cocaine relapse [16,19,20,21].

## 2. Materials and Methods

### 2.1. Animal Housing and Catheter Implantation

Male Wistar rats (225–250 g; Charles River Laboratories, Sulzfeld, Germany) were individually housed in standard rodent cages, except those from abstinence in an enrichment environment, where they were kept 3–4 per cage (see below). All experiments took place during a light phase 12:12 h light/dark cycle in a temperature- and humidity-controlled environment. Animals had free access to water, while food (VRF1 pellets, UK) was given ad libitum, except at initial lever pressing and prior to the beginning of self-administration periods, where it was mildly limited (ca. 24 g/rat/day). The rats, after a week-long acclimatization and initial lever pressing (see below), were implanted with back-mounted cannulas fixed to a silastic catheter inserted into the right jugular vein under anaesthesia, as described previously in detail by Wydra et al., 2015) [22]. In the first three days postsurgery, meloxicam (Metacam, Boehringer Ingelheim; 5 mg/kg, *s.c*.) was used to reduce postoperative pain. Following surgery, all animals had a 7-to-9-day recovery period during which the catheters were flushed daily with 0.2 mL of a sterile 0.9% NaCl solution containing ceftriaxone (Polpharma, Poland; 100 mg/mL) and heparin sodium (Polfa S.A., Poland; 100 IU/mL) to preserve the catheters’ patency and to prevent infection. The procedures were carried out in accordance with the European Union Directive (2010/63/EU) and with the approval of the local ethics committee (approval number 144/2020). The number of animals was limited, according to the principle of “3Rs”.

### 2.2. Drugs

Cocaine hydrochloride (Toronto Research Chemicals (TRC), Toronto, ON, Canada) and esketamine hydrochloride (Tocris, UK) were dissolved in sterile 0.9% NaCl and administered through *i.v.* (in a volume of 0.1 mL per infusion) or *i.p.* (in a volume of 1 mL/kg). The esketamine doses were selected based on previous publications [8,23] showing the drug’s effectiveness after single injections in different behavioural tasks in rodents.

### 2.3. Behavioural Procedures

#### 2.3.1. Initial Lever Press Training

Rats were trained two hours daily to press an “active” lever in standard operant chambers (Med Associates Inc., St. Albans, VT, USA) with a fixed ratio (FR) 1 food (sweetened milk, Gostyń, Poland) reinforcement. After meeting the criteria (100 rewards/2 h session), the FR was subsequently increased to 5.

#### 2.3.2. Cocaine Self-Administration Procedure

After 7–9 days of recovery, the rats began the self-administration procedure in the operant chambers (above). The self-administration of cocaine (2 h daily sessions performed 6 days/week) occurred on a FR 5 schedule, where presses on the “active” lever resulted in a single cocaine infusion (0.5 mg/kg/infusion in 0.1 mL), accompanied by a 5 s presentation of a cue-conditioned stimuli (light and tone, 2000 Hz; 15 dB). Following each cocaine infusion, there was a 20 s timeout. The house lights switched on throughout each session. Presses on the “inactive” lever were recorded from the 4th day of cocaine self-administration and this lever was not reinforced. The acquisition of the conditioned operant response lasted 14 days until the animals met a stable number of “active” lever presses during 3 continuous sessions [19]; only these rats were directed to abstinence conditions and reinstatement tests.

#### 2.3.3. Cocaine Abstinence Procedures

##### Extinction Training Exposure

After the cocaine self-administration, the animals (two separate groups with *n* = 5–6 rats/group) underwent a 10-day extinction training in the experimental cage, where saline was delivered and there was no presentation of the conditioned stimulus during the 2 h daily sessions.

##### Social Isolation

After the cocaine self-administration, the animals (two separate groups, *n* = 7 rats/group) underwent a 10-day social isolation. Rats lived individually in an isolation cage, which was a plastic cage with white walls and were cleaned approximately once weekly to reduce social interactivity.

##### Enriched Environment Exposure

After the cocaine-self-administration, the animals (two separate groups, *n* = 7 rats/group) underwent abstinence in an enriched environment. Three–four animals were housed in larger cages equipped with several hiding places and toys of different materials (plastic, wood, cotton), which were replaced 3 times per week to maintain novelty. The rats were handled several times per day.

#### 2.3.4. Reinstatement of Cocaine-Seeking Behaviour

Following cocaine abstinence, the rats were tested for a response reinstatement induced either through cocaine prime (10 mg/kg; *i.p.*) or cue-induced reinstatement (light and tone previously associated with cocaine self-administration) in 2 h daily sessions. During the cocaine-induced reinstatement, a response on the “active” lever resulted in the *i.v.* saline infusion. During the cue-induced reinstatement, a response on the “active” lever was reinforced with a 5 s light + tone presentation, together with a saline infusion. During the reinstatement tests, responses on the “inactive” lever had no programmed consequence. The rats were pretreated with esketamine or a vehicle 60 min before the reinstatement sessions. Each rat underwent only one type of reinstatement procedure, with a maximum of four tests.

### 2.4. Locomotor Activity

Locomotor activity, defined here as a distance travelled, was measured in Opto-Varimex cages (Columbus Instruments, Columbus, OH, USA), as described previously by Wydra et al. [24]. Drug-naïve and nonhabituated animals (*n* = 8 rats/group) were injected with esketamine or a vehicle 60 min before the start of the 2 h individual locomotor recordings.

### 2.5. Statistical Analyses

All data (the mean ± SEM) were performed with Statistica 13.3 (TIBCO Software Inc., Palo Alto, CA, USA) and visualized in GraphPad Prism 9.4.1 (GraphPad Software, Inc., San Diego, CA, USA). The normality of distributions and the equal homogeneity of variance were checked with the Shapiro–Wilk test and Levene test, respectively. For normally distributed data, a one-way repeated measure ANOVA (factors: “active” lever/“inactive” lever + treatment; F statistic) or a two-way repeated measure ANOVA (factors: “active” lever + “inactive” lever + treatment; F statistic) were performed. When the data were found not to be normally distributed, a nonparametric Kruskal–Wallis test with ranks (H statistic) or the Friedman test (for “active” or “inactive” lever, respectively; Q statistic) were employed. In significant differences between groups, either parametric Dunnett’s or nonparametric Dunn’s multiple comparison *post hoc* tests were performed. Comparisons were considered significant with a *p* value < 0.05.

## 3. Results

### 3.1. Cocaine Self-Administration

The rats (*n* = 39) met the criterion of cocaine self-administration. The mean of the total cocaine intake was >140 mg/rat (Table 1). For the extinction training abstinence conditions (Experiment 1), the cocaine-self-administering rats pressed the “active” lever more frequently than the “inactive” lever from the fourth cocaine self-administration session to the first extinction day (Q_(3, 11)_ = 16.88, *p* < 0.0001; Figure 1). Similarly, for the social isolation (Experiment 2) and enriched environment (Experiment 3) abstinence conditions, the rats pressed the “active” lever more frequently than the “inactive” lever from the fourth to the fourteenth cocaine self-administration session (Q_(3, 11)_ = 18.73, *p* < 0.0001 and Q_(3, 11)_ = 16.91, *p* < 0.0001, respectively).

### 3.2. Cocaine Abstinence Procedures

After the cocaine self-administration, the rats (Table 1) were tested for the reinstatement of drug-seeking induced through either cocaine prime (10 mg/kg, *i.p*., Group 1) or the presentation of the drug-associated cue (Group 2) in combination with esketamine or a vehicle.

#### 3.2.1. Experiment 1—Impact of Esketamine on Cocaine-Seeking Behaviour following Extinction Training Exposure

After the extinction training in both the cocaine prime- and cue-induced seeking behaviour groups, the rats pressed 85% and 65% more frequently on the “active” lever as compared to the last day of extinction training, respectively (Figure 2A,B).

Only in a dose of 10 mg/kg did esketamine show a potent and statistically significant decrease in “active” lever presses toward the cocaine-induced reinstatement condition (Q_(4, 6)_ = 13.40, *p* = 0.0006) without changes in “inactive” lever presses (Q_(4, 6)_ = 0.25, *p* = 0.9761; Figure 2A). Likewise, after the presentation of the drug-associated cue, the rats showed a significant main effect of the treatment (F_(3, 24)_ = 5.91, *p* = 0.0036) and lever (F_(1, 8)_ = 33.67, *p* = 0.0004), as well as an interaction effect (F_(3, 24)_ = 3.94, *p* = 0.0202; Figure 2B). Here, esketamine in both doses of 5 and 10 mg/kg produced a significant reduction in “active” lever presses amounting to 46% and 57%, respectively (Figure 2B).

#### 3.2.2. Experiment 2—Impact of Esketamine on Cocaine-Seeking Behaviour following Social Isolation

After abstinence with social isolation, esketamine in doses of 5 and 10 mg/kg evoked a reduction (by 68% and 72%, respectively (Q_(3, 7)_ = 11.14, *p* = 0.0012; Figure 3A)) in “active” lever presses linked with cocaine prime in rats. A significant inhibitory effect of a separate dose of esketamine (5 mg/kg) was observed (F_(2, 12)_ = 5.42, *p* = 0.0210; Figure 3A) for the “inactive” lever presses (Figure 3A).

Similarly, for the cue-induced reinstatement condition, the rats showed a main effect of the treatment (F_(2, 24)_ = 17.31, *p* < 0.0001), lever (F_(1, 12)_ = 16.65, *p* = 0.0015) and interaction effect (F_(2, 24)_ = 5.32, *p* = 0.0122; Figure 3B). Esketamine in doses of 5 and 10 mg/kg reduced the number of “active” lever presses by 56% and 69%, respectively, as compared to the vehicle-pretreated animals (Figure 3B).

#### 3.2.3. Experiment 3–Impact of Esketamine on Cocaine-Seeking Behaviour following Enrichment Environment Exposure

After enrichment environment abstinence, the pretreatment with esketamine showed a main effect of the treatment (F_(3, 36)_ = 3.96, *p* = 0.0154), lever (F_(1, 12)_ = 24.58, *p* = 0.0003) and interaction effect (F_(3, 36)_ = 3.09, *p* = 0.0391) in the rats that underwent the cocaine-induced reinstatement (Figure 4A). Only in a dose of 10 mg/kg did esketamine reduce the cocaine-induced reinstatement, showing as a 64% decrease in the number of “active” lever presses compared to the vehicle pretreatment (Figure 4A). For the cue-induced cocaine-seeking behaviour, the pretreatment with 2.5 or 5 mg/kg of esketamine suppressed the number of “active” lever presses by 58 and 68% (Q_(4, 7)_ = 15.26, *p* = 0.0016), respectively, as compared to the pretreatment with the vehicle (Figure 4B). For this type of reinforcement, no significant effect was observed for “inactive” lever presses between the groups (Q_(4, 7)_ = 5.08, *p* = 0.1655; Figure 4B).

### 3.3. Effects of Esketamine on Locomotor Activity

No alteration in locomotor activity (H_(3, N = 32)_ = 1.09; *p* = 0.777) was observed after acutely administering esketamine (5–15 mg/kg) in rats (Figure 5).

## 4. Discussion

There are no efficacious medication treatments for CUD, while psychosocial activities, named cognitive–behavioural therapy or contingency management, have been evidenced to have many benefits, including a reduction in the relapse risk [21,25]. Several former literatures, including the results from our laboratory, support the evidence of the impact of environmental conditions [16,19,26] and for its control mechanisms [19,27,28,29] on cocaine-seeking behaviour. Shortly, an enriched environment or extinction training decreased the risk or rate of relapse, while social isolation resulted in an increase [17,18]. Here, using pharmacological analyses, we reported that esketamine reduced cocaine seeking induced through either drug-associated cues or cocaine priming.

Interestingly, the diminishing actions of esketamine on the cocaine-induced reinstatement were independent of withdrawal conditions as being evident, following extinction training, social isolation or exposure to an enriched environment. Further, they were specific because esketamine in an effective dose range towards cocaine seeking neither modified the “inactive” level presses (except from 5 mg/kg of esketamine in animals exposed to abstinence with social isolation before cocaine priming) in cocaine-experienced rats with a different history of withdrawal nor locomotor activity in drug-naïve animals. However, some limitations existed here, as long-term cocaine exposure, despite no effects on “inactive” lever presses, might have affected the baseline activity of the rats. The inhibitory action of esketamine in a lower (5 mg/kg), but not higher (10 mg/kg), dose was unexpected, even more so that some locomotor inhibitory effects were noticed for esketamine doses (15 mg/kg) in the vehicle-pretreated rats, but not repeatedly adrenocorticotropic-hormone (ACTH)-treated rats [23], while other authors recently demonstrated a lack of motor effects of esketamine in a dose range of 15–20 mg/kg in rats [30,31]. Although a significant effect, the number of inactive lever presses following the 5 mg of esketamine was similar to the response during the cocaine self-administration or vehicle-pretreated rats during the cocaine reinstatement test in rats forwarded to abstinence with social isolation.

A possible mechanism by which esketamine may work within cocaine seeking may include: (i) treating depressive symptoms; (ii) extending the withdrawal time; (iii) blocking the reconsolidation of drug-related memories; (iv) provoking mystical experience with enhancing “psychological” therapy efficacy; and/or (v) substitution for the subjective effects of cocaine.

Cocaine withdrawal activates the stress systems and several withdrawal-associated symptoms have been noted in humans and laboratory animals. They include—among others—depression and anxiety. In fact, esketamine has shown antidepressant effects in preclinical research [8,9] and clinical trials [10,11], which may have had significant implications in reducing the risk of relapse to cocaine in rats in the present study. Additionally, a very recent paper by Yang et al. (2022) demonstrated that esketamine alleviated anxiety-like behaviours in a rat model of post-traumatic stress disorder [12]. Similarly, in male rats submitted to a panicogenic challenge, esketamine caused the rapid-onset of anxiolytic-like effects [32]. Such mood-enhancing properties could be tied with esketamine’s reduction in cocaine seeking. On the other hand, it should be remembered that exposure to an enriched environment during withdrawal helped the rats to adapt rapidly to potential stress effects, and such animals coped better with stress than the rats housed in isolation, which was also evidenced by a decrease in the plasma corticosterone levels observed in the enriched rats in comparison to the isolated rats, following the cocaine self-administration [33]. This observation, together with the similar inhibitory effects of esketamine in all groups of rats, seemed to limit the mood enhancement of the latter drug in the studied cocaine behavioural outcome. The restoration of drug-related memories should also be discussed in the context of the inhibition of cocaine seeking through the use of esketamine, since cocaine withdrawal is characterized by poor focus and slowed thoughts in humans [34] and with the impairment of working memory in rats [35,36,37,38]. In human research, an impairment in cognition was associated with acute ketamine administration, but recent studies demonstrated a small-to-moderate loss of cognition (including verbal learning and memory figures) after acute racemic ketamine [39], while esketamine caused a much lower decline in concentration capacity and primary memory [40] in healthy volunteers. Similarly, in individuals with treatment-resistant depression, esketamine did not evoke significant deleterious neurocognitive effects [41]. On the other hand, a separate report in rats indicated that esketamine could alleviate surgery-induced postoperative cognitive decline assessed in the open field, novel object recognition or Morris’s water maze tests [42], which may have partly counteracted cocaine seeking in the present paper.

This issue, as well as the potential impact of the psychotomimetic side effects of esketamine that might have affected the experience of the animals, requires further investigations. Thus far, clinical findings have shown transient perceptual disturbances of ketamine [43,44], while laboratory data indicate psychomimetic-like effects of ketamine in rats [45] and mice [46]. Such effects might be linked with esketamine-induced increases in DA release [7] and reduced DA D2/3 receptor binding in the striatum [13].

When considering the inhibitory effects of esketamine, the possibility that this drug could substitute for cocaine’s subjective effects and, thus, reduce the response to cocaine should also be taken into account. In fact, both cocaine and ketamine produced reinforcement in self-administration procedures in monkeys; however, they differed in their effects on hypothalamic–pituitary–adrenal axis activity. While cocaine stimulated, ketamine inhibited ACTH and cortisol secretion, which speaks to the little commonality between these drugs [47]. On the other hand, it should also be remembered that esketamine (systemic administration)—as well as ketamine—increased Glu release in the rodent prefrontal cortex (PFC) [48]. In the case of ketamine, only the systemic drug injection increased Glu in the medial PFC, while the local injection did not affect the Glu release [48]. The origin of the ketamine effect on mPFC Glu is unknown. The local (intra-PFC) esketamine effects on the Glu release were not determined; however, ketamine through inhibiting cortical NMDA receptors expressed in γ-aminobutyric acid (GABA) neurons preferentially evoked the disinhibition of GABA interneurons, enhancing the firing of pyramidal neurons [49]. The enhancement of the firing of pyramidal neurons and Glu transmission from the PFC to the nucleus accumbens core has been linked with seeking and relapse behaviours, while withdrawal from chronic cocaine has been associated with reduced basal accumbal core Glu levels [50]. Using MR spectroscopy in human cocaine addicts, Glu levels in the PFC reduced during cocaine self-administration, while cocaine withdrawal either normalized it or returned it to precocaine Glu levels [51]. Whether esketamine, through the enhancement of the Glu release in the PFC, can mimic cocaine challenge doses given to cocaine-sensitizing animals [52] or cocaine priming in reinstatement tests [53], further studies are strongly requested. In this context, it is also urgently needed to elucidate the effects of esketamine in cocaine addicts, since the only available data in human research have indicated a significant anticocaine craving action induced through ketamine [54], while another recent double-blinded randomized–controlled trial did not indicate the efficacy of ketamine in a long-term treatment [55].

The complexity of the pharmacological mechanism of the action of esketamine was not investigated here, and remains unanswered. However, several possibilities should be taken into account: (1) the primary mode of the action of cocaine is the indirect increase in the extracellular level of DA [56] and thought activation; however, the psychostimulant also modulates glutamate release from corticostriatal terminal DA D_2_ receptors [57]; (2) cocaine seeking and relapse are mediated through experience-built synaptic plasticity and the rise in the incentive motivational value of cocaine through which the neurochemical background is stored within glutamate signalling [16,26,58,59,60]; (3) cocaine withdrawal reduces basal glutamate levels [61]; and (4) changes in glutamate ionotropic NMDA receptor subunit composition transit from the occasional use of cocaine to CUD [62].

Interestingly, using the same behavioural procedures we recently documented increased the levels of the GluN1 and/or GluN2A subunit of NMDA receptors in rats exposed to cocaine abstinence with extinction training in the nucleus accumbens [28], prelimbic cortex and dorsal hippocampus [19]. Further, there were region-specific changes in both the expression of the GluN2B subunit and NMDA receptor trafficking during different cocaine abstinence conditions [19]. Thus, an increase in the GluN2B expression in the total homogenate from the dorsal hippocampus was found under an enriched environment and isolation. On the other hand, cocaine abstinence with social isolation increased the GluN2B and GluN2B/PSD95 complex levels in the PSD fraction of the prelimbic cortex, while the GluN2B-receptor-selective antagonist CP 101,606 parallel attenuated cue-induced cocaine-seeking behaviour and diminished neurochemical correlates in the rat [19]. The role of the ventral hippocampal GluN2B subunit in the attenuation of cocaine-seeking behaviour was reported with the intravenous Tat-NR2B9c peptide, which disrupted the interaction of PSD95 with GluN2B, following the extinguishing of cocaine operant responses [29,63]. An interesting overlap between the present study and previous research examining the role of withdrawal conditions was the recent RNA-seq analysis showing the robust transcriptomic differences within several pathways in the nucleus accumbens shell of rats exposed to different withdrawal conditions [64]. In fact, animals in an enrichment environment had—among others—glutamate receptor signalling (e.g., Grin2d that codes NMDA epsilon-4 subunit in humans) upregulated, while the isolation condition downregulated the activity of these pathways [65]. The above-cited findings may also speak for esketamine’s actions on the GluN2 subunit toward reducing cocaine-seeking behaviour. In fact, esketamine has a high binding affinity for NMDA receptors [66]; however, it is—probably like ketamine—not selective for the specific GLUN2A-2D subunits of the NMDA receptor. It is unclear which of these subunits are responsible for the reported behavioural effects.

Despite few clinical [67,68] and preclinical present data being optimistic, further studies would better clarify the numerous unknowns related to the use of esketamine in CUD and clarify the mechanisms by which ketamine reduces cocaine seeking. The most helpful seem to be genetic or optogenetic tools used to parse out the contributions of NMDA receptors and/or DA system functioning in the behavioural effects of esketamine on cocaine seeking.

## 5. Conclusions

The development of drug treatments directed at brain Glu signalling (including the antagonism of NMDA receptors) is attractive as a method for finding potential antiaddiction medications. Our behavioural and pharmacological studies showed the efficacy of esketamine in reducing cocaine seeking in rats in different abstinence conditions.

## Figures and Tables

**Figure 1 biomolecules-13-01411-f001:**
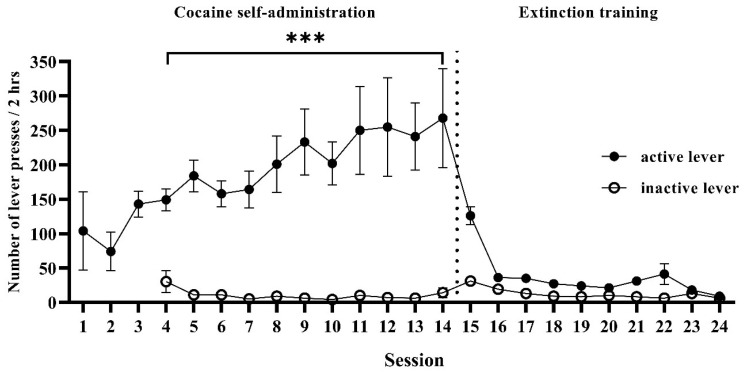
Cocaine self-administration and extinction training in rats. Numbers of “active” (black dots) and “inactive” (white dots) lever presses during 2 h sessions. *n* = 11 rats. Data were expressed as mean ± SEM. Data were comparable between levers (*** *p* < 0.001; Friedman’s test, followed by Dunn’s multiple comparison test).

**Figure 2 biomolecules-13-01411-f002:**
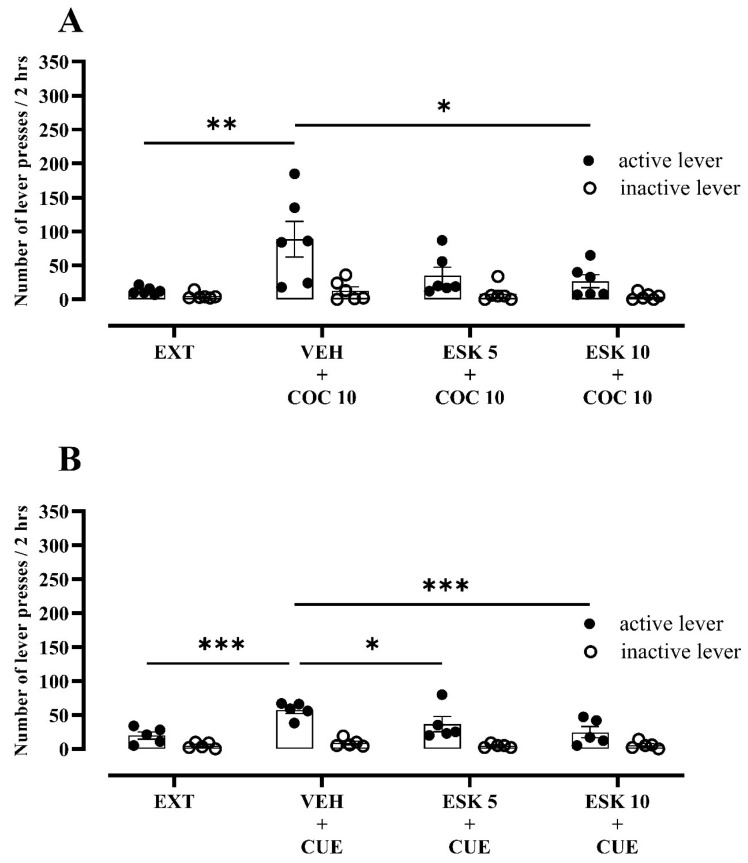
Effects of esketamine (ESK, 5–10 mg/kg) on (**A**) cocaine prime- (COC, 10 mg/kg) or (**B**) CUE (light and tone previously associated with cocaine self-administration)-induced reinstatement in rats that underwent cocaine abstinence with extinction training (EXT) from cocaine self-administration. *n* = 6 rats/COC group and 5 rats/CUE group. Data were expressed as mean ± SEM. Data were comparable to vehicle (VEH) + COC 10 or VEH + CUE group (* *p* < 0.05; ** *p* < 0.01; *** *p* < 0.001; two-way repeated measure ANOVA or Friedman’s tests, followed by Dunnett’s or Dunn’s multiple comparison tests, respectively).

**Figure 3 biomolecules-13-01411-f003:**
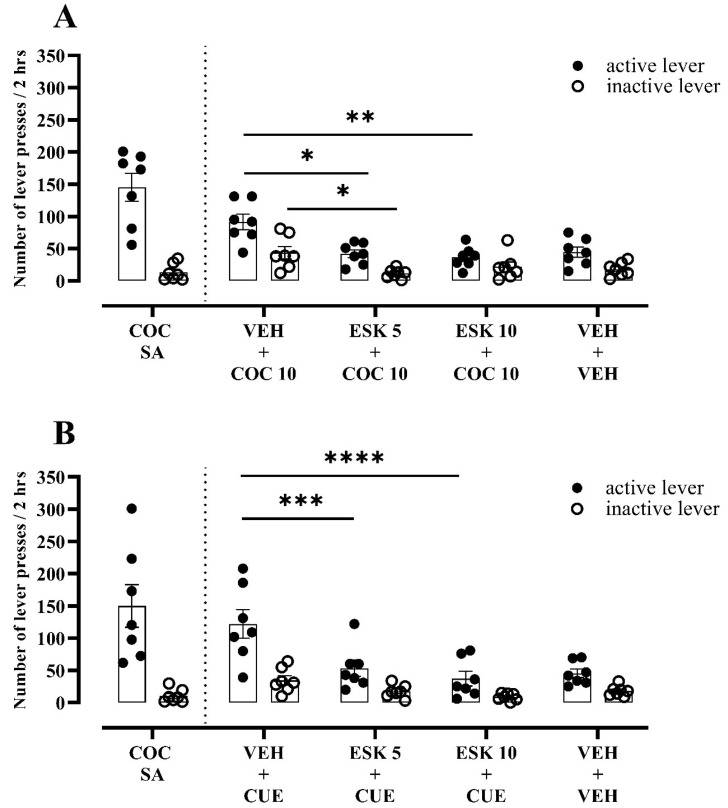
Effects of esketamine (ESK, 5–10 mg/kg) on (**A**) cocaine prime- (COC, 10 mg/kg) or (**B**) CUE (light and tone previously associated with cocaine self-administration)-induced reinforcement in rats that underwent cocaine abstinence with social isolation from cocaine self-administration (COC SA). *n* = 7 rats/group. Data were expressed as mean ± SEM. Data were comparable to vehicle (VEH) + COC 10 or VEH + CUE group (* *p* < 0.05; ** *p* < 0.01; *** *p* < 0.001; **** *p* < 0.0001; two-way repeated measure ANOVA or Friedman’s tests, followed by Dunnett’s or Dunn’s multiple comparison tests, respectively).

**Figure 4 biomolecules-13-01411-f004:**
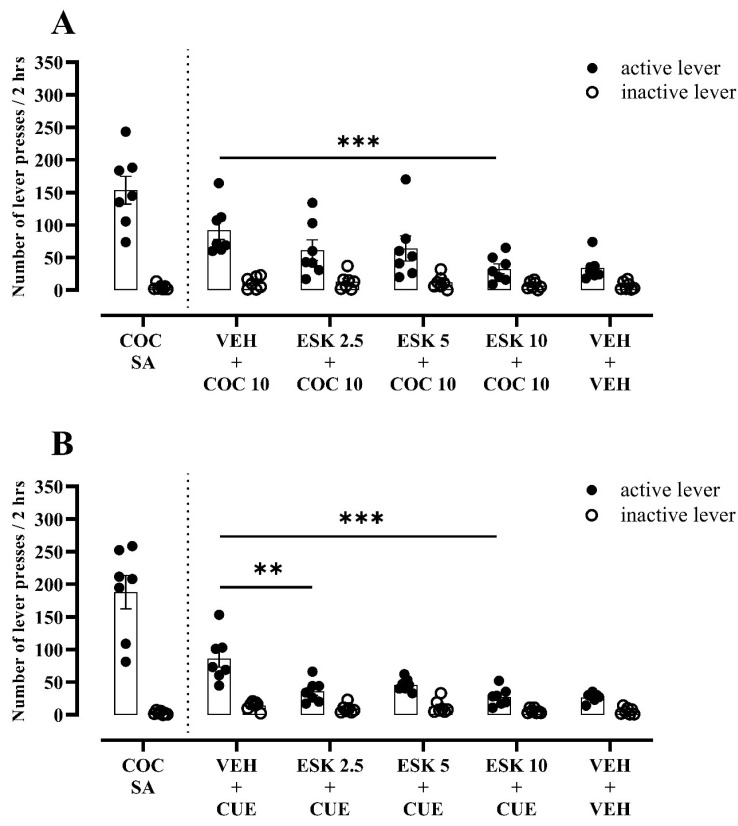
Effects of esketamine (ESK, 2.5–10 mg/kg) on (**A**) cocaine prime- (COC, 10 mg/kg) or (**B**) CUE (light and tone previously associated with cocaine self-administration)-induced reinstatement in rats that underwent cocaine abstinence in enrichment environment from cocaine self-administration (COC SA). *n* = 7 rats/group. Data were expressed as mean ± SEM. Data were comparable to vehicle (VEH) + COC 10 or VEH + CUE group (** *p* < 0.01; *** *p* < 0.001; two-way repeated measure ANOVA or Friedman’s tests, followed by Dunnett’s or Dunn’s multiple comparison tests, respectively).

**Figure 5 biomolecules-13-01411-f005:**
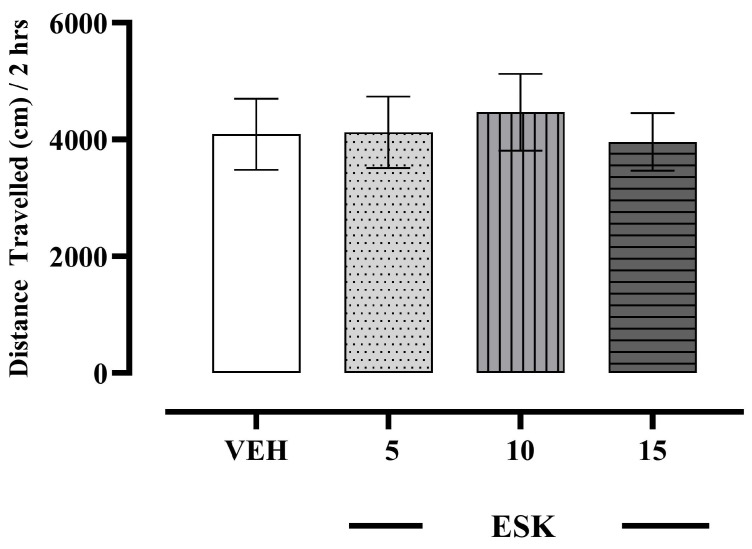
Effects of esketamine (ESK, 5–15 mg/kg) on locomotor activity, expressed as distance travelled (cm) measured during 2 h recordings. *n* = 8 rats/group. Data were expressed as mean ± SEM. Data were comparable to vehicle (VEH) group (Kruskal–Wallis test).

**Table 1 biomolecules-13-01411-t001:** Cocaine (0.5 mg/kg/infusion) self-administration in rats. The number of “active” and “inactive” lever presses and cocaine infusions during the last 3 self-administration sessions in rats, as well as the total cocaine intake over 14 days of cocaine self-administration. Data were expressed as mean ± SEM.

Cocaine Abstinence Conditions		Cocaine Self-Administration			
		Number of “Active” Lever Presses	Number of “Inactive” Lever Presses	Number of Drug Infusions	Total Cocaine Intake (mg/rat)
Experiment 1 Extinction training	Group 1	237 ± 28	7 ± 3	35 ± 3	197 ± 19
	Group 2	174 ± 27	9 ± 8	31 ± 6	175 ± 31
Experiment 2 Social isolation	Group 1	146 ± 21	14 ± 5	26 ± 4	161 ± 21
	Group 2	150 ± 33	10 ± 4	24 ± 5	141 ± 20
Experiment 3 Enrichment environment	Group 1	153 ± 21	5 ± 2	26 ± 3	147 ± 18
	Group 2	188 ± 26	3 ± 1	29 ± 4	158 ± 27

## Data Availability

All data related to this manuscript are available upon request.

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
