# Peer review of "Esketamine Inhibits Cocaine-Seeking Behaviour Subsequent to Various Abstinence Conditions in Rats"

_biomolecules, 2023, doi:10.3390/biom13091411_

Round 1

Reviewer 1 Report

The authors have successfully demonstrated the inhibiting effects of esketamine on cocaine-seeking behavior reinstatement in rats that underwent different abstinence conditions. They have also suggested that this effect of esketamine might be independent of changes in locomotor activity. A few possible pharmacological mechanisms were also discussed by the authors, which needed further investigation. The complexity of the design of their experiment is commendable and clearly supported their hypothesis. However, improvements can still be made in some aspects to make their work more appropriate for publication.

Major comments:

Introduction

1. While the authors did mention the mechanism of actions of cocaine and ketamine, the rationale of using esketamine as a potential agent for reducing drug-seeking behavior was not mentioned. What specific neurobiological factors that esketamine can modulate which are also involved in cocaine addiction and relapse? Simply stating that both cocaine and esketamine affect dopamine/glutamate levels may imply a multitude of possible effects on drug-seeking behavior, given also that ketamine influences several neuronal systems.

2. How would the fact that there is still a growing concern regarding the safety and abuse potential of esketamine influence its use as a potential treatment for addiction?

Discussion

1.  It was mentioned by the authors that different abstinence conditions may affect drug-seeking reinstatement (line 65-69 and 257-258). However, this was not evident in the results (VEH + COC 10, VEH + CUE, and/or VEH + VEH groups vs. pre-abstinence) and must also be provided with an explanation, taking into account potential confounding factors in their experimental protocol.

2.       Although the authors provided potential pharmacological and molecular mechanisms for the effect of esketamine, which likewise warrant further investigation, their theories might be more supported had they provided a few evidence for such mechanisms. (i) The authors suggested that it might be through the effect of esketamine on the depressive/withdrawal symptoms during cocaine abstinence that caused a reduction in cocaine reinstatement. The authors may then have demonstrated related behaviors in rats abstaining from cocaine before and after esketamine treatment. (ii) The authors also mentioned that it might be through glutamate mediation of memory and the blockage of drug-related memory reconsolidation during abstinence that esketamine may have exhibited its effect in cocaine reinstatement. Determining glutamate levels or glutamate receptor expression in abstaining rats before and after esketamine treatment would have potentially implicated such mechanisms in its effect.

Minor comments:

Introduction

1. Please rephrase lines 31-33 since the thought is unclear. Also, do the authors mean “glutamatergic” instead of “glutaminergic”?

2.  Line 45: remove “the”

3.  Line 60: citation [22] must be placed at the end of the statement.

4.  Line 62: remove “the drug”

5. It was mentioned that AMPA activation may occur following NMDA antagonism (line 39-40). How would this potentially affect drug-seeking behavior and reinstatement?

Materials and methods

1. How did the authors ensure that catheters will not be removed from rats during exposure to the enriched environment, given that 3-4 rats are housed together?

2.  Were there no limits to the training day for rats that never reached 100 rewards at all?

3.  “Social isolation” might be a more appropriate term than “home cage exposure”, since rats were all probably exposed to their “home cages”.

4.  Line 121: please correct “weakly”

5.  Citation in line 141 is missing.

Results

1.  What was the reason for not showing the inactive lever presses in the 1st three days of cocaine SA (Figure 1)?

2.  In figure 3, 5 mg/kg esketamine also reduced inactive lever pressing in socially isolated rats that were primed with cocaine, suggesting that cocaine priming may have also increased inactive lever pressing in socially isolated rats. A brief explanation can be provided since this finding was already mentioned by the authors (line 265-266).

3. Any particular reason for using an additional dose (2.5 mg/kg) in Experiment 3 (Figure 4)?

4. For locomotor activity (Figure 5), would it not be possible to confirm the effect of esketamine on locomotion in cocaine-exposed and abstaining rats, as the baseline activity of rats may have already been compromised by long-term cocaine exposure?

Some statements need rephrasing to make the thought clearer.

Author Response

Reviewer 1

We appreciate a lot the comments of this Reviewer on our ms. We have addressed all Reviewer’s remarks which improved our paper.

Comments and Suggestions for Authors

The authors have successfully demonstrated the inhibiting effects of esketamine on cocaine-seeking behavior reinstatement in rats that underwent different abstinence conditions. They have also suggested that this effect of esketamine might be independent of changes in locomotor activity. A few possible pharmacological mechanisms were also discussed by the authors, which needed further investigation. The complexity of the design of their experiment is commendable and clearly supported their hypothesis. However, improvements can still be made in some aspects to make their work more appropriate for publication.

Major comments:

Introduction

  1. While the authors did mention the mechanism of actions of cocaine and ketamine, the rationale of using esketamine as a potential agent for reducing drug-seeking behavior was not mentioned. What specific neurobiological factors that esketamine can modulate which are also involved in cocaine addiction and relapse? Simply stating that both cocaine and esketamine affect dopamine/glutamate levels may imply a multitude of possible effects on drug-seeking behavior, given also that ketamine influences several neuronal systems.

We have modified the Introduction to better clarify choosing esketamine in our research (the revised version: page 1, lines 27-31 and 33-44; page 2, lines 49-51). Also, the possibility that esketamine might substitute for cocaine, and therefore decreased drug-seeking has been discussed (the revised version: page 11, lines 306-308).

  1. How would the fact that there is still a growing concern regarding the safety and abuse potential of esketamine influence its use as a potential treatment for addiction?

We agree with the Reviewer’s reservation about the limitation of esketamine use. The discussion and conclusion have been modified. 

Discussion

  1. It was mentioned by the authors that different abstinence conditions may affect drug-seeking reinstatement (line 65-69 and 257-258). However, this was not evident in the results (VEH + COC 10, VEH + CUE, and/or VEH + VEH groups vs. pre-abstinence) and must also be provided with an explanation, taking into account potential confounding factors in their experimental protocol.
  2. Although the authors provided potential pharmacological and molecular mechanisms for the effect of esketamine, which likewise warrant further investigation, their theories might be more supported had they provided a few evidence for such mechanisms. (i) The authors suggested that it might be through the effect of esketamine on the depressive/withdrawal symptoms during cocaine abstinence that caused a reduction in cocaine reinstatement. The authors may then have demonstrated related behaviors in rats abstaining from cocaine before and after esketamine treatment. (ii) The authors also mentioned that it might be through glutamate mediation of memory and the blockage of drug-related memory reconsolidation during abstinence that esketamine may have exhibited its effect in cocaine reinstatement. Determining glutamate levels or glutamate receptor expression in abstaining rats before and after esketamine treatment would have potentially implicated such mechanisms in its effect.

We fully agree that further studies are needed to estimate the mechanisms of esketamine actions, including determination of protein or gene expression for glutamate receptors in rats before and after esketamine treatment. Future research directions have been added to the Discussion (the revised version: page 12, lines 356-367).

Minor comments:

Introduction

  1. Please rephrase lines 31-33 since the thought is unclear. Also, do the authors mean “glutamatergic” instead of “glutaminergic”?

We have changed these sentences and corrected the word „glutamatergic” (the revised version: page 1, lines 30-31).

  1.  Line 45: remove “the”

            We have changed these sentences.

  1. Line 60: citation [22] must be placed at the end of the statement.

The position of citation 22 has been replaced (the revised version: page 2,  line 68).

  1. Line 62: remove “the drug”

We have corrected it and changed a few sentences (the revised version: page 1, lines 27-31; 33-44; and page 2, lines 49-51).

  1. It was mentioned that AMPA activation may occur following NMDA antagonism (line 39-40). How would this potentially affect drug-seeking behavior and reinstatement?

In fact, AMPA activation in the PFC induces drug-seeking behavior and reinstatement (e.g., Cornish and  Kalivas, J Neurosci. 2000). However, the esketamine’s actions as AMPA receptor activator are not evident so far (they are reported for ketamine only). According to the suggestion of Reviewer 2 the paragraph related to description of ketamine’s neurochemistry was deleted from the Introduction (the revised version: page 1, lines 33-44). So, in the revised version we have not discussed AMPA mechanisms of esketamine.

Materials and methods

  1. How did the authors ensure that catheters will not be removed from rats during exposure to the enriched environment, given that 3-4 rats are housed together?

            We have used dummy cannulae to protect the guide cannulae from damage. Every day during experiments we have controlled the placement of dummy cannula and replaced them if damaged or lost.

  1. Were there no limits to the training day for rats that never reached 100 rewards at all?

            In this experiment, all our animals reached the criteria of 100 rewards during a

            2 hr session. So there were no extra training days for rats.

  1. “Social isolation” might be a more appropriate term than “home cage exposure”, since rats were all probably exposed to their “home cages”.

We fully agree with the Reviewer, that the phrase „home cage exposure” has been replaced by „social isolation”.

  1. Line 121: please correct “weakly”

We have changed it (the revised version: page 3, line 110).

  1. Citation in line 141 is missing.

We have added this citation (the revised version: page: 3,  lines: 130-131).

Results

  1. What was the reason for not showing the inactive lever presses in the 1st three days of cocaine SA (Figure 1)?

The detailed procedure was designed by Smaga et al., 2021. In fact, during the first 3 days of cocaine self-administration, the „inactive lever” was upsent. This information has been added to the manuscript (the revised version: page 3, lines: 96-97).

  1. In figure 3, 5 mg/kg esketamine also reduced inactive lever pressing in socially isolated rats that were primed with cocaine, suggesting that cocaine priming may have also increased inactive lever pressing in socially isolated rats. A brief explanation can be provided since this finding was already mentioned by the authors (lines 265-266).

The discussion of the inhibitory effects of esketamine 5 mg/kg toward inactive lever has been added (the revised version: page 10, lines 256-258).

  1. Any particular reason for using an additional dose (2.5 mg/kg) in Experiment 3 (Figure 4)?

To get dose-response effects for the inhibitory action of esketamine, we used this lower dose of esketamine.

  1. For locomotor activity (Figure 5), would it not be possible to confirm the effect of esketamine on locomotion in cocaine-exposed and abstaining rats, as the baseline activity of rats may have already been compromised by long-term cocaine exposure?

It is a very good suggestion. However, for the revision time of the ms it was impossible to perform such a study. We have indicated it as a limitation in the Discussion (the revision version: page 10, lines 256-258).

We have corrected the editing and improved quality of the English language in the manuscript.

Yours sincerely,

Małgorzata Filip, Professor, DSci., PhD

Department of Drug Addiction Pharmacology

Head of Maj Institute of Pharmacology, Polish Academy of Sciences

Smetna 12, 31-343 Kraków, Poland

mal.fil@if-pan.krakow.pl

&

Karolina Wydra, PhD

Department of Drug Addiction Pharmacology

Maj Institute of Pharmacology, Polish Academy of Sciences

Smetna 12, 31-343 Kraków, Poland

wydra@if-pan.krakow.pl

Reviewer 2 Report

This is an interesting basic research study designed to examine the potential use of esketamine to treat cocaine use disorder.  The paper is well written (though some attention to word use and grammar are needed), and the methods and results clear.  Also, there are some font issues in the title and through the manuscript that need to be fixed.

The main interpretive issue for me (that the authors touch on) is the degree to which esketamine is antagonizing the effects of cocaine, rather than substituting for cocaine's subjective effects, and thus reducing responding for cocaine.

Minor Comments:

Lines 12-14: I do not quite understand this sentence.

Lines 14-15: Can delete the sentence about the racemic mixture and add this information after 'esketamine' on line 15.

Line 19: locomotor activity

Line 29-30: 'important'

Line 31: Delete 'significantly'

Line 141: Reference is missing

Lines 206-207: It is stated that there is a significant effect on the inactive lever presses, though p>0.1.

Very good English with only a few issues with word choice.

Author Response

#Reviewer 2

We would like to thank the Reviewer for the important remarks to our manuscript. All issues raised by the Reviewer have been taken into consideration and the ms has been modified.

This is an interesting basic research study designed to examine the potential use of esketamine to treat cocaine use disorder.  The paper is well written (though some attention to word use and grammar are needed), and the methods and results clear.  Also, there are some font issues in the title and through the manuscript that need to be fixed.

The main interpretive issue for me (that the authors touch on) is the degree to which esketamine is antagonizing the effects of cocaine, rather than substituting for cocaine's subjective effects, and thus reducing response for cocaine.

It is a very important remark. This potential mechanism of esketamine has been added to the Discussion (the revised version: page 10, lines 256-266, and page 11, lines 304-334).

Minor Comments:

Lines 12-14: I do not quite understand this sentence.

We have changed it (the revised version: page 1, lines 12-14).

Lines 14-15: Can delete the sentence about the racemic mixture and add this information after 'esketamine' on line 15.

We have deleted these sentences and changed the next ones (the revised version: page 1, line 12-14)

Line 19: Locomotor activity

We have corrected the phrase (the revised version: page 1, line 18).

Line 29-30: 'Important'

Accordingly to this and the second reviewer, we have changed the sentences (the revised version: page 1, lines 27-31).

Line 31: Delete 'significantly'

We have deleted it and changed the sentence (the revised version: page 1, lines 30-31)

Line 141: Reference is missing

We have added a reference (the revised version: page 3, lines 130-131)

Lines 206-207: It is stated that there is a significant effect on the inactive lever presses, though p>0.1.

We have corrected it (the revised version: page 5, line 195)

We have corrected the editing and improved the English language quality in the manuscript.

Yours sincerely,

Małgorzata Filip, Professor, DSci., PhD

Department of Drug Addiction Pharmacology

Head of Maj Institute of Pharmacology, Polish Academy of Sciences

Smetna 12, 31-343 Kraków, Poland

mal.fil@if-pan.krakow.pl

&

Karolina Wydra, PhD

Department of Drug Addiction Pharmacology

Maj Institute of Pharmacology, Polish Academy of Sciences

Smetna 12, 31-343 Kraków, Poland

wydra@if-pan.krakow.pl

Round 2

Reviewer 1 Report

Job well done to the authors for revising their manuscript. All concerns were addressed. While it was acknowledged that the manuscript had limitations, stating them would indeed direct future studies and other improvements regarding this work.

Only minor corrections in English and wordings are needed. Please review the following lines: 

30-31: "...strong impact of the glutamatergic signaling over control cocaine seeking..."

49-50: "...Esketamine was chosen sine it possess...propertyes that can partly tied with reduction..."

307: remove "the" 

310-311: "...cortisol secret what speak about a little commonality between..."

367: remove "here"

Author Response

Again, we really acknowledge the Reviewer input on our manuscript.

All indicated language mistakes have been corrected.

Comments and Suggestions for Authors

Job well done to the authors for revising their manuscript. All concerns were addressed. While it was acknowledged that the manuscript had limitations, stating them would indeed direct future studies and other improvements regarding this work.

Comments on the Quality of English Language

Only minor corrections in English and wordings are needed. Please review the following lines: 

30-31: "...strong impact of the glutamatergic signaling over control cocaine seeking..."

Corrected.

49-50: "...Esketamine was chosen sine it possess...propertyes that can partly tied with reduction..."

Corrected.

307: remove "the" 

Corrected.

310-311: "...cortisol secret what speak about a little commonality between..."

Corrected.

367: remove "here"

Corrected.

We have corrected the editing and improved quality of the English language in the manuscript.

Yours sincerely,

Małgorzata Filip, Professor, DSci., PhD

Department of Drug Addiction Pharmacology

Head of Maj Institute of Pharmacology, Polish Academy of Sciences

Smetna 12, 31-343 Kraków, Poland

mal.fil@if-pan.krakow.pl

&

Karolina Wydra, PhD

Department of Drug Addiction Pharmacology

Maj Institute of Pharmacology, Polish Academy of Sciences

Smetna 12, 31-343 Kraków, Poland

wydra@if-pan.krakow.pl